# Self-compassion and the impostor phenomenon: Associations and implications

Kristýna Krejčová[1]*, Lucie Kvasničková Stanislavská[1], Fabio Ibrahim[2]

**1** Department of Management and Marketing, Faculty of Economics and Management, Czech University of Life Sciences Prague, Prague, Czech Republic, **2** Department of Personality Psychology and Psychological Assessment, Faculty of Humanities and Social Sciences, Helmut Schmidt University, Hamburg, Germany

* krejcovak@pef.czu.cz

## Abstract

Impostor phenomenon reflects persistent self-doubt and fear of failure despite success. This study examined the association of impostor phenomenon with self-compassion in 601 Czech university students (368 women; $M_{age}$ = 23.46, $SD$ = 3.05), using the Impostor-Profile 30 and Self-Compassion Scale. Employing correlation analysis, regression, and structural equation modelling, we investigated the relationships among the constructs and their dimensions. In the correlation analysis, competence doubt was moderately associated with isolation ($r$ = .409) and overidentification ($r$ = .427). The regression analysis indicated these subscales as being significantly associated with competence doubt ($R^2$ = .26, F(6, 594) = 35.18, p < .001). Structural equation modelling further supported the model with stronger links between negative self-compassion dimensions and impostor phenomenon compared with positive ones (e.g., self-kindness) (CFI = .99, TLI = .99, RMSEA = .031). These findings highlight the importance of considering feelings of isolation and overidentification when designing or interpreting self-compassion-focused approaches in the context of impostor phenomenon, particularly among university student populations vulnerable to stress and low self-efficacy.

## Introduction

Impostor Phenomenon (IP) describes the long-term and complex psychological state of unrealistic self-perceptions accompanied by symptoms of anxiety. Individuals with IP experience permanent feelings of inauthenticity and a fear of failure being exposed [1]. This state is far more severe than occasionally perceived inadequacies or self-doubt [2]. Although IP is not included in the Diagnostic and Statistical Manual of Mental Disorders or the International Classification of Diseases, its impact on general mental health and well-being is significant. Employees with IP are more vulnerable to avoidant coping strategies and emotional exhaustion, which may lead to job

**Data availability statement:** All de-iden-tified data supporting the results of this study have been deposited in the Mendeley Data Repository under DOI: 10.17632/n85kvhm6mn.1.

**Funding:** The author(s) received no specific funding for this work.

**Competing interests:** The authors have declared that no competing interests exist.

dissatisfaction [3]. Therefore, IP is correlated with the incidence of burnout syndrome and compassion fatigue [4].

Self-compassion (SC) is a multifaceted self-attitude that reflects how individuals internally respond to experiences of failure and inadequacy. [5]. In contrast to the impostor phenomenon (IP), which reflects predominantly negative and performance-contingent self-evaluation with specific cognitive-emotional patterns, SC encompasses a broader range of self-reflective responses, and its operationalisation includes both compassionate and uncompassionate elements. The next key distinction between IP and SC is the extent to which they reflect reality. While SC attitudes are derived from the mindful awareness of the present experience, IP is influenced by an unrealistic distortion of self-attitudes [6].

Despite their apparent multidimensional connections, the relationship between SC and the IP has been under-researched. However, "self-compassion is likely a buffer against the effects of IP" [7] and may help an individual step out of the vicious circle of IP-related thoughts [2]. Liu et al. [8] revealed a significantly stronger relationship of IP with negative SC components than positive SC components. Patzak et al. [9] arrived at similar conclusions and identified the role of gender in the possible protective effect of SC on IP. Although the existing literature offers valuable findings, the relationship between IP and SC requires further research.

The present study addressed these gaps by examining the complex interactions between IP and SC, observing correlations of positive and negative SC dimensions with IP components. Considering the demographic factors of impostorism and the growing incidence of mental health problems among university students [10], we focused on the population of students in their final year of bachelor's and master's degrees at the Faculty of Economics and Management (FEM), Czech University of Life Sciences (CZU). These students usually possess initial work experience, which helps examine the interaction between IP and SC in the context of career transition. The selection of this research population aligns with studies that reflect the high occurrence of IP in a competitive academic environment [11] and demonstrates the negative impact of IP on career planning and striving among university students [12].

Our findings provide specific insights into the associations between IP and SC dimensions and suggest perspectives for SC-focused approaches for individuals beginning of their careers. Furthermore, our results may contribute to the development of university strategies for promoting well-being, which is crucial for effective education [13].

### Research background

**Impostor phenomenon.** The existence of IP was first detected by Clance and Imes [14] in a sample of highly functioning women, who were unable to experience feelings of objective success. They suffered from the vicious circle of the dread of failure, anxiety symptoms, over-preparation and subsequent fraud, and feelings of self-doubt after success, which they were unable to internalize. Later research suggested that IP is observed regardless of gender, ethnicity, or age [15].

IP was revealed as a predictor of anxiety, depression, and suicidal thoughts [16] and identified as a comorbidity factor with anxiety, depression, low self-esteem, and

social dysfunction [15]. Moreover, the impostor's self-attitude mediates a negative correlation between perfectionism and happiness [17] and a positive relationship between perfectionism and depression [18]. Considering the environmental influences that nurture feelings of IP, studies have highlighted competitiveness and achievement pressures in the academic environment [11,12]. The impostor phenomenon is a multidimensional construct, comprising the following core elements: the impostor cycle, the need to be special or the very best, superman or superwoman aspects, a fear of success, the denial of competence and discounting praise, and fear and guilt about success [14]. According to Clance, individuals who fulfil at least two of these criteria are classified as experiencing IP [14]. The psychometric assessment of these core elements is crucial for capturing the nuanced nature of the construct and for supporting the diagnostic decisions based on Clance's original conceptualization. In a meta-analysis, Mak et al. [19] criticized the existing instruments assessing IP for failing to adequately reflect its multidimensional structure. Nevertheless, the recent research extended the perception of the Clance Impostor Phenomenon Scale as a multidimensional tool encompassing not only the total score but also its three facets [20].

Based on the discrepancy perceived at that time between the theoretical construct and its psychometric operationalization, Impostor-Profile 30 (IPP 30) was developed. It captures the facets of IP through six subscales and one total score, enabling a differentiated and global assessment of the phenomenon. The competence doubt scale concerns self-doubts and a fear of failure, the working style scale measures tendencies toward procrastination or precrastination, alienation targets the feeling of authenticity, and other-self divergence addresses how respondents perceive the expectations of others. The frugality scale is related to self-expectation, and the need for sympathy refers to the importance of appearing sympathetic [21].

The IPP30 conceptualises impostorism as a multidimensional profile with distinct facets, which aligns more closely with Clance's original description of a constellation of impostor-related tendencies. This facet-level view allows domain-specific theorising and practical targeting (e.g., identifying which components of impostorism are most relevant for intervention), and it enables profile or typology approaches that cannot be derived from a single total score.

**Self-compassion.** As defined by Neff [5], SC involves dimensions of "(a) self-kindness—extending kindness and understanding to oneself rather than harsh judgment and self-criticism, (b) common humanity—seeing one's experiences as part of the larger human experience rather than seeing them as separating and isolating, and (c) mindfulness— holding one's painful thoughts and feelings in balanced awareness rather than over-identifying with them" [5]. Despite the tendency to have a dualistic view of SC versus compassion for others [22], both concepts are interrelated and have a cumulative effect on well-being [23].

The defined dimensions form the operationalised structure of the SC. Considering the specific relationships between the single subscales of the Self-Compassion Scale (SCS), Dreisoerner et al. [24] hypothesized a link between self-kindness, common humanity and isolation. This reflected a vicious circle of self-criticism leading to shame, less socially oriented behavior and social exclusion. In their experiment, mindfulness training led to an increase in total SC and self-kindness and a decrease in isolation. The training in common humanity was associated with lower over-identification and higher SC and life satisfaction.

SC has been associated with various indicators of well-being. A higher SC score leads to a better balance in reflecting one's needs against the needs of others [25], improves emotional self-regulation based on the realistic acceptance of negative emotions [5], decreases stress levels by creating a protective buffer against stressors [26], positive affection [27] and better mental health [28]. Accordingly, SC promotes work-related well-being in terms of job satisfaction [29] and resilience [30], and reduces burnout symptoms [31]. Furthermore, it mediates the relationship between work-related stress and the incidence of anxiety and depressive symptoms from a long-term perspective [26,32]. Although the level of SC is related to one's personality structure [33], it can be enhanced through SC-centred and mindfulness-oriented interventions [34]. These interventions promote anchoring in the present moment, which helps counter excessive self-criticism. By fostering reality-based self-perceptions, these interventions can counteract the underestimation of abilities, which is characteristic of the IP [7].

**IP and SC: Well-being-related contexts and differences.** IP and SC have been widely studied in relation to well-being and other psychological outcomes.. The impact of IP on well-being and motivation is generally self-handicapping based on unrealistic perceptions of one's achievement [2]. By contrast, SC supports general well-being, resilience, and job satisfaction [29,30] but does not contribute to self-boosting. Its effect stems from self-acceptance in times of failure, facilitated by a realistic perception of its causes and significance [35]. Furthermore, SC does not involve attitudes of self-centeredness, passivity, and self-pity, as it prevents overidentification with one's own problems by fostering perceptions from a broader perspective as an inherent part of human existence [5].

The nature of the mutual relationships between self-attitudes associated with IP and SC becomes more apparent when examining their connections to self-efficacy— a key concept in self-reflective processes. Self-compassion encompasses cognitively and emotionally anchored beliefs about the ability to reach specific goals, influence motivation and persistence in pursuing them, and supports striving to achieve them despite obstacles and frustrations [36]. Unlike SC, IP is negatively associated with self-efficacy. This relationship highlights the paradoxical vicious circle of impostorism. Individuals with IP experience maladaptive perfectionism [4] and belong to high-achieving families [37]. Therefore, we expect them to have high aspirations and commitment in reaching their goals. Self-efficacy positively influences academic persistence. However, the self-efficacy of individuals with high IP is generally low [17]. By contrast, meta-analysis of the relationship between SC and self-efficacy revealed a positive relationship, which indicates a possible improvement in self-efficacy after SC-oriented interventions [38].

Although the relationship with self-efficacy is central to differentiating SC and IP, these constructs also show opposing relationships with other motivation- and well-being-related concepts. Emotional exhaustion and burnout are positively correlated with IP [3,4,39] but can be prevented with the help of SC [40]. While IP stimulates avoidance coping [3], SC supports the acceptance of negative emotions as a necessary condition for successful processing [5]. Neuroticism as a personality trait is negatively related to SC [41] but positively related to IP [32]. This partial finding supports the general opposing relationship of IP and SC with well-being, emphasizing the negative influence of the IP [3,17] and the positive influence of SC [42].

Although IP and SC are conceptually multidimensional and potentially interrelated, research that directly investigates their relationship is limited [7–9,43]. Based on previous findings, our study aimed to identify the mutual relationships between SC and the IP measured by the SCS [35,44] and IPP30 [21]. Using the IPP 30 to analyse the relationship between SC and IP is an innovative element of the research design because the findings of previous studies [7–9,43,45] are based on the more traditional Clance Impostor Phenomenon Scale [14] or short Impostorism scale [46]. Furthermore, a correlation analysis and path analysis specified the mutual relationships between the IP and SC dimensions.

Hence, we hypothesized the following:

H1: Internal correlations within the IPP subscales and within the SCS subscales are stronger than those between the IPP and SCS subscales.

H2: Higher scores on the positive SCS subscales (self-kindness, common humanity, and mindfulness) are associated with lower scores on the IPP 30.

H3: Higher scores on the negative SCS subscales (self-judgment, isolation, and over-identification) are associated with higher scores on IPP 30.

## Materials and methods

### Procedure

The recruitment period for the research was conducted between October 29 and December 13, 2024. After removing the incomplete responses, we obtained a sample of 601 respondents aged between 21 and 38 years ($M = 23.46$, $SD = 3.05$) All participants were bachelor's and master's students of the Faculty of Economics and Management (FEM), Czech

University of Life Sciences (CZU). They were recruited using a non-probability convenience sampling strategy, primarily through university online platforms. Participation was voluntary, and no financial incentives were provided. The sex ratio was slightly imbalanced, with 368 women (61.2%) and 232 men (1 respondent preferred not to answer), which roughly corresponded to the sex structure of FEM students. For the complex SEM models, samples >200 are recommended [47], whereas AMOS models should be based on larger samples than partial least squares path modelling [48,49]. Our sample met these criteria, and the recommendation based on five times the number of respondents than the size of the model indicators for covariance-based models [49]. If we considered the items of the questionnaires as indicators, we would have required 280 respondents; this limit was significantly exceeded. The data were collected using *Formr*, an open-source survey framework [50]

This study was approved by the Ethics Committee of the CZU, Prague (File reference number 4/2024). It was conducted in accordance with the ethical principles of the Declaration of Helsinki. All the respondents provided written informed consent to participate in the study. The data were analyzed anonymously in accordance with the ethical standards of the American Psychological Association.

## Measures

**Impostor-Profile 30.** The IPP 30 [21] comprises 30 items across six subscales (competence doubt, working style, alienation, other-self divergence, frugality, and need for sympathy) as well as a total score. The responses are measured on a 10-point Likert scale, ranging from 1 (*Not like me at all*) to 10 (*Very much like me*). The subscales demonstrate good internal consistency, with Cronbach's alpha values ranging from.94 to.72 [51]. One exception is the need for sympathy scale, which shows lower reliability with an α = .67 [21]. Following the guidelines for cross-cultural adaptation, the questionnaire was translated using a forward–backward translation procedure [52]. The first translator translated the items from English into Czech. A second independent translator then translated the Czech version back into English. Both English versions were verified by the original author to confirm equivalence.

**Self-Compassion Scale.** The SCS is a broadly adapted instrument with 26 items measuring six subscales and a total score [35,44]. The subscales correspond to the basic SC dimensions as defined by Neff [5]. The internal reliability of the subscales (Cronbach's α between.75 and.81) and total score (Cronbach's α = .92) has been established [44,53]. The convergent, discriminant, predictive, and known-groups validity have been assessed as good [35]. Nevertheless, some studies doubt the construct validity because of an unclear factor structure or suggest different factor solutions [54–58]. Therefore, we only took separate scores from the subscales in the analysis, which is in line with the recommendations by Neff [44] and was used in many other studies [53,59–61] The official Czech adaptation of the full-length version was used [55].

In summary, the selected instruments were chosen due to their solid psychometric foundation and their availability in professionally translated Czech versions with established validation parameters [55,62], as well as because they reflect the theoretical conceptualisation of the examined phenomena as operationalised in the present study.

**Analytic plan.** The initial statistical analysis was performed using IBM SPSS, Version 30. In addition to the descriptive statistics, our analysis tested the normality of the distribution using the Kolmogorov–Smirnov test, which is a non-parametric test of the normality of the distribution. Based on the results, the correlation coefficient was selected for further analysis [63]. Additionally, linear regression was performed to test the SCS subscales as the predictors and the IPP subscales as the dependent variables.

Considering the literature review [36] and the correlation analysis outputs, a theoretical model was created and tested for consistency with our data in IBM SPSS Amos, Version 28. The path analysis was conducted using the Maximum Likelihood (ML) estimation method, which is the default estimator in this software. To model fit, we considered the following widely used indexes: comparative fit index (CFI) and Tucker–Lewis index (TLI), which compare the fit of a hypothesized model with the baseline model, and root mean square error of approximation (RMSEA), which indicates the difference

between a hypothesized model and perfect model [64,65]. The following criteria were reflected: CFI and TLI with values close to.95 indicated superior fit, RMSEA<.05 indicated good fit, RMSEA<.08, indicated reasonable errors of approximation in the population, RMSEA.08–.10, indicated mediocre fit, and RMSEA>.10, indicated poor fit [64]. Hypothesis 1 was tested using a correlation analysis of the direction, strength, and significance of the relationships between the SCS and IPP 30 subscales. Hypotheses 2 and 3 were tested using linear regression, with the positive and negative SCS subscales to examine the variance in the IPP 30 subscales. Based on the theoretical background and outputs of conventional data analysis, the hypothesized path-analysis model of complex relationships between the SCS and IPP 30 subscales was specified and tested according to the selected criteria. Based on the theoretical background, we built on the relationships between self-kindness, common humanity and isolation in the SCS, as described by Dreisoerner et al. [24], and the dominant correlation between isolation and overidentification, as reflected by Cleare et al. [61].

## Results

Table 1 summarizes the descriptive statistics for IPP 30 and SCS. In correspondence with Neff [66], the mean SCS scores were assessed according to the following criteria: a mean score of 1–2.5 is low, 2.5–3.5 is moderate, and 3.5–5 is high (row "Mean/Item"). In this framework, all subscales' outputs are considered moderate, with slightly higher values for positive subscales.

Standard deviations indicated adequate variability with all values well within commonly accepted thresholds for normality. Skewness and kurtosis values suggested no substantial departures from normality. Skewness ranged from −0.569 to 0.350 and kurtosis from −0.739 to 0.377, indicating only slight asymmetry and relatively flat distributions.

To examine potential gender effects, independent-samples $t$-tests were conducted. Women scored significantly higher than men on isolation, overidentification, competence doubt and need for sympathy, whereas men scored significantly higher on mindfulness and frugality (see Table 2).

Normality was assessed using the Shapiro–Wilk test. Although several significant deviations from normality were observed, these were expected given the large sample size and do not invalidate the use of parametric tests, as skewness and kurtosis values remained within acceptable ranges.

**Table 1. Descriptive statistics for the IPP 30 and SCS.**

| Variable | Mean | Median | SD | M/Item |
|---|---|---|---|---|
| CD | 60.89 | 62.00 | 17.81 | x |
| WS | 35.61 | 36.00 | 12.6 | x |
| A | 13.07 | 13.00 | 5.73 | x |
| O | 19.73 | 20.00 | 6.87 | x |
| F | 18.6 | 19.00 | 5.72 | x |
| NS | 19.4 | 20.00 | 5.82 | x |
| SK | 14.56 | 14.00 | 4.19 | 2.91 |
| SJ | 15.23 | 15.00 | 3.76 | 3.05 |
| CH | 12.03 | 12.00 | 3.54 | 3.00 |
| I | 12.83 | 13.00 | 3.72 | 3.21 |
| M | 12.56 | 12.00 | 2.92 | 3.14 |
| OI | 13.32 | 13.00 | 3.17 | 3.33 |

Abbreviations: IPP-30 (Impostor-Profile 30), SCS (Self-Compassion Scale), CD (competence doubt), WS (working style), A (alienation), O (other-self divergence), F (frugality), NS (need for sympathy), SK (self-kindness), SJ (self-judgment), CH (common humanity), I (isolation), M (mindfulness), OI (overidentification), x = not assessed, M/Item = Mean/Item

 

**Table 2. Gender differences – Descriptive statistics.**

| Variable | Mean (M) | SD (M) | Mean (W) | SD (W) | Levene's p | t-test p |
|---|---|---|---|---|---|---|
| SK | 14.53 | 3.887 | 14.57 | 4.377 | .033 | .906 |
| SJ | 14.97 | 3.786 | 15.40 | 3.743 | .575 | .168 |
| CH | 12.36 | 3.649 | 11.83 | 3.451 | .626 | .073 |
| I | 12.08 | 3.912 | 13.30 | 3.522 | .110 | <.001 |
| M | 13.35 | 2.764 | 12.06 | 2.912 | .830 | <.001 |
| OI | 12.47 | 3.207 | 13.86 | 3.035 | .417 | <.001 |
| CD | 55.69 | 17.820 | 64.19 | 17.035 | .132 | <.001 |
| WS | 34.80 | 12.235 | 36.17 | 12.810 | .430 | .194 |
| A | 13.48 | 6.152 | 12.82 | 5.450 | .064 | .165 |
| O | 20.00 | 7.016 | 19.54 | 6.785 | .554 | .430 |
| F | 19.44 | 6.001 | 18.08 | 5.487 | .204 | .005 |
| NS | 18.68 | 5.933 | 19.88 | 5.700 | .147 | .014 |

Abbreviations: M(men), W(women), CD (competence doubt),WS (working style), A (alienation), O (other-self divergence), F (frugality), NS (need for sympathy), SK (self-kindness), SJ (self-judgment), CH (common humanity), I (isolation), M (mindfulness), OI (overidentification).

Note. For each variable, Levene's test was used to determine the appropriate t-test; only the relevant row is reported.

The correlation analysis revealed the expected relationships between the individual scales of both questionnaires. The strongest associations were detected between the subscales in frame of the same dimensions, namely positive and negative SC dimensions. Nevertheless, significant moderate associations were observed between the IP subscale of competence doubt and SCS negative subscales of isolation and overidentification. These associations were stronger than the internal relationships between the IP subscales and between the majority of SCS subscales (and higher than relationships between positive and negative SCS subscales) (Table 3). Therefore, we rejected H1.

Based on the correlation analysis results, a simple linear regression was performed with the competence doubt subscale as the dependent variable and the SC subscales as the predictors. The regression model was significant, $F_{(6,594)}$ = 35.18, p<.001, with an $R^2$ = .26, indicating that the SC subscales were associated with 26% of the variance in competence doubt. The three associations were statistically significant: Overidentification ($b$ = 1.34, standard error (SE) = 0.26, $t$ = 5.16, p<.001, Isolation ($b$ = 1.04, SE = 0.21, $t$ = 4.88, p<.001) and Common Humanity ($b$ = −.597, SE = 0.22, $t$ = −2.75, p = .006). The interpretation of 95% confidence intervals further supported these findings, indicating that CH, I, and OI were statistically significantly associated with the outcome, as their intervals did not include zero, whereas those for SK, SJ, and M did (Table 4). Variance inflation factors (VIFs) ranged from 1.41 to 1.76, indicating no multicollinearity concerns.

Among the remaining subscales, the regression model predicting Need for Sympathy was not statistically significant, whereas all other models reached statistical significance (p values ranging from <.001 to.037). At the level of individual regression coefficients, a significant association was observed only between Alienation and Isolation ($b$ = .428, SE = 0.77, $t$ = 5.52, p<.001, 95% CI [0.275, 0.580]), as well as between Working Style and Overidentification ($b$ = .531, SE = 0.207, $t$ = 2.57, p = .011, 95% CI [0.125, 0.938]), and between Other-Self Divergence and Isolation ($b$ = .275, SE = 0.94, $t$ = 2.91, p = .004, 95% CI [0.09, 0.460]). Based on the regression analysis, considering competence doubt as the dependent variable, we reject H2, detecting only a single significant association of IP subscale with the positive SCS subscale, but not H3.

In the path analysis, we specified directional paths, interpreted as statistical associations rather than causal effects. We modelled the statistical pathways from isolation, overidentification, self-kindness, and common humanity to competence doubt. Common humanity was included in an indirect statistical pathway linking self-kindness and competence doubt.t. Overidentification and Isolation showed a direct association with competence doubt. Moreover, competence doubt was

**Table 3. Correlation matrix (Pearson's r).**

|     | CD | WS | A | O | F | NS | SK | SJ | CH | I | M | OI |
|-----|-----|-----|-----|-----|-----|-----|-----|-----|-----|-----|-----|-----|
| CD | 1 | 0.381** | 0.262** | 0.358** | −0.033 | 0.150** | −0.281** | 0.270** | −0.226** | 0.409** | −0.309** | 0.427** |
| WS |   | 1 | 0.012 | 0.145** | −0.072 | 0.180** | −0.184** | 0.164** | −0.141** | 0.151** | −0.173** | 0.206** |
| A |   |   | 1 | 0.324** | −0.020 | −0.148** | −0.020 | 0.089* | 0.003 | 0.243** | −0.072 | 0.102* |
| O |   |   |   | 1 | 0.119** | −0.004 | −0.063 | 0.077 | −0.033 | 0.178** | −0.060 | 0.141** |
| F |   |   |   |   | 1 | 0.313** | 0.081* | 0.013 | 0.079 | −0.039 | 0.131** | −0.076 |
| NS |   |   |   |   |   | 1 | −0.062 | 0.099* | −0.028 | 0.098* | −0.055 | 0.126** |
| SK |   |   |   |   |   |   | 1 | −0.343** | 0.516** | −0.258** | 0.525** | −0.236** |
| SJ |   |   |   |   |   |   |   | 1 | −0.102* | 0.424** | −0.202** | 0.435** |
| CH |   |   |   |   |   |   |   |   | 1 | −0.076 | 0.466** | −0.109** |
| I |   |   |   |   |   |   |   |   |   | 1 | −0.287** | 0.574** |
| M |   |   |   |   |   |   |   |   |   |   | 1 | −0.378** |
| OI |   |   |   |   |   |   |   |   |   |   |   | 1 |

Note: $p < .05$ (*), $p < .01$ (**).

Abbreviations: CD (competence doubt), WS (working style), A (alienation), O (other-self divergence), F (frugality), NS (need for sympathy), SK (self-kindness), SJ (self-judgment), CH (common humanity), I (isolation), M (mindfulness), OI (overidentification).

**Table 4. Simple linear regression for Competence doubt and SCS.**

|     | Unstandardized coefficients | | Standardized coefficients | t | Sig. | 95% CI | |
|-----|-----|-----|-----|-----|-----|-----|-----|
|     | B | Std. error | Beta |   |   | Lower bound | Upper bound |
| SK | −.280 | .199 | −.066 | −1.410 | .159 | −.671 | .110 |
| SJ | .125 | .198 | .026 | .630 | .529 | −.265 | .514 |
| CH | −.597 | .217 | −.119 | −2.748 | .006 | −1.023 | −.170 |
| I | 1.041 | .214 | .218 | 4.875 | <.001 | .622 | 1.461 |
| M | −.369 | .279 | −.060 | −1.323 | .186 | −.916 | .178 |
| OI | 1.343 | .260 | .239 | 5.161 | <.001 | .832 | 1.854 |

Abbreviations: SK (self-kindness), SJ (self-judgment), CH (common humanity), I (isolation), M (mindfulness), OI (overidentification)

associated with working style and other-self divergence. The covariance between self-kindness, overidentification and isolation was also included in the model specification (Fig 1).

The proposed model fit the data well. The $\chi^2/df$ ratio was 1.57, which fell below the commonly accepted threshold of 3, indicating an adequate model fit [47]. Both the CFI (=.99) and TLI (=.99) exceeded the criteria of 0.95 for superior fit. The RMSEA (=.031) also met the requirements for a good fit (<.05), suggesting that the difference between the observed and hypothesized population covariance matrices, adjusted for model complexity, was low [64]. Furthermore, the Akaike information criterion value of 64.78, which was lower than that of the saturated model (70.00), indicated a perfect balance between the goodness of fit and model parsimony. All observed indices confirmed that the hypothesized model corresponded well with the observed data. After establishing that the model fit the data well, we examined the structural relationships among the constructs. Following Cohen's classification [67], most of the path coefficients in our model were assessed as small or medium in terms of effect strength; some of them exceeded the limit for strong associations.

Isolation demonstrated a positive path to competence doubt in terms of a small-to-moderate positive relationship ($\beta = .24, p < .001$). Overidentification showed a moderate positive association with the competence doubt ($\beta = .27, p < .001$).

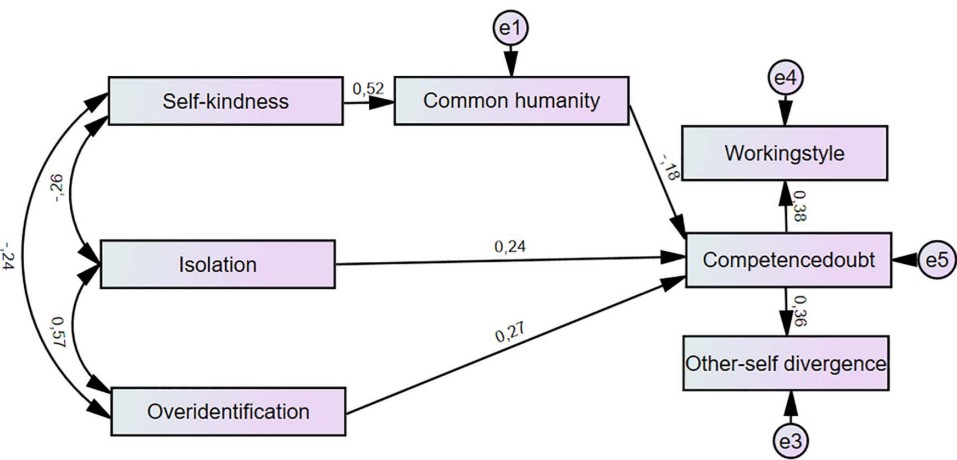

**Fig 1. Path analysis model of the selected variables.**

Competence doubt showed a moderate positive relationship with other-self divergence *(β = .36, p < .001)* and working style *(β = .38, p < .001)*. The association between Isolation and Overidentification was positive *(β = .57, p < .001)*. Self-kindness was negatively associated with Isolation (β = −.26, p < .001), Overidentification (β = −.24, p < .001), and strongly and positively related to common humanity *(β = .52, p < .001)*. The negative association between common humanity and competence doubt was the weakest in the model *(β = −.18, p < .001)*.

## Discussion

### Summary of results

Our study focused on the relationships between the components of SC and IP, measured using the SCS and IPP30, respectively. Despite the psychometric validation of the SCS, including the total score [35,44], our research revealed marginal correlations between the positive and negative SCS subscales in the correlation analysis and path analysis of complex relationships. This finding is consistent with the two-factor solution of the SCS suggested in some studies [54,57,58].

In the correlation analysis, the relationship between the IP subscale of competence doubt and the negative SC subscales of isolation and overidentification was higher than the internal correlations between the IP subscales and most internal correlations between the SC subscales. Although the observed correlations were generally small to moderate in magnitude, the relatively large sample size increases confidence in the stability of these associations. Thus, our study completely corresponds with that of Liu et al., who identified a significantly stronger relationship between IP and negative SC than that between IP and positive SC [8]. This observation is also referred to in other studies [9,43] and mirrors the general relationship between SC and psychopathological symptoms, which are also more intense in the case of negative SC subscales [68].

The significance of the association detected in correlational analysis was also evident from the complex analysis of relationships in the path-analysis model. These relationships highlighted in the model are consistent with the regression outputs, including the estimated coefficients, their statistical significance, and the associated confidence intervals. In particular, competence doubt showed significant associations with common humanity, isolation, and overidentification. Although associations between competence doubt and the self-kindness/mindfulness subscales were significant in the correlational analysis, these effects were not retained in the regression model, suggesting that shared variance among conceptually related predictors may be involved.

In conclusion, our research contributes to the understanding of the complex relationships between IP and SC in several ways and emphasizes the possible buffer effect of SC-focused interventions in targeting impostor feelings. However, causal claims cannot be derived based on the cross-sectional design of our study. The contributions of our study can be considered from various perspectives. First, we examined these constructs in a large sample of Czech university students (N = 601), thereby extending the evidence base beyond the predominantly US or Western European contexts. Second, some studies did not use specific SC scales in relation to Impostor Phenomenon [8].

Furthermore, the previous researchers focused on IP-SC relationships [7–9] using the Clance Impostor Phenomenon Scale [14]. This scale has traditionally been treated as unidimensional, despite ambiguous empirical evidence. Recent research, however, supported a bifactorial model comprising three group factors in addition to a general IP factor [20]. Nevertheless, IPP30 with six facets, allows for the observation of more nuanced relationships between IP and SCS. [10]. Our findings indicated that feelings of isolation and overidentification were particularly relevant to impostorism, especially competence doubt, alienation, working style and other-self divergence. Competence doubt captures the core self-evaluative uncertainty and fear of failure that is most proximal to self-compassion processes. Isolation and overidentification reflect two mechanisms that plausibly intensify these doubts: social disconnection (I am the only one struggling) and cognitive fusion with negative self-evaluations (ruminative overinvolvement). Consistent with this rationale, competence doubt showed the strongest and most consistent associations with isolation and overidentification in our data, motivating its central role in the regression and path models.

Generally, our facet-level results suggest that the link between self-compassion and impostorism is not uniform but concentrated in competence doubts and the negative self-compassion dimensions. This pattern refines buffer accounts by highlighting isolation and overidentification as potential processes associated with competence-related impostor doubts.

## Theoretical and practical implications

The specific relationships between the SC and IP dimensions identified in our model provide valuable insights for IP prevention and intervention, which is consistent with the current research. Generally, SC supports crucial mental health indicators such as resilience and well-being [28,30,69,70]. The level of SC can be enhanced via targeted SC-centered and mindfulness interventions [34,71,72]. Moreover, SC supports the nonjudgmental acceptance of negative emotions [5,73]. Consequently, it soothes neurotic symptoms and avoidant behaviour [5,41], typically noticed in individuals with IP [3,32]. Furthermore, SC training boosts self-efficacy [38], a personal construct negatively affected by IP [17].

Although our results are not as straightforward as those of Patzak [9], who did not identify a correlation between the IP measure and the positive SC subscale of common humanity, the associations with the positive SC subscales were non-significant compared with the negative ones. In the context of these findings, a meta-analysis of SC-related interventions highlighted a more substantial effect for the negative SC subscales than for the positive ones; however, this output was not supported with statistical comparisons [72]. These findings may be partially attributed to the negative wording of the items in the impostor questionnaire. Nevertheless, this does not reduce the empirical significance of the results. Furthermore, the findings illustrated that the positive and negative SC subscales are not complementary and, therefore, should not be treated as opposites [74]. In conclusion, efficient preventive strategies and interventions for IP should reflect the negative SC subscales and related personality factors to alleviate doubts about competence, which impact other impostor self-attitudes.

The association of the overidentification subscale with competence doubt and working style converges with previous findings related to IP. For instance, an experimental study investigating the relationship between IP and failure attribution showed that individuals with high impostor tendencies tend to attribute failure internally and stably, whereas success is attributed externally and unstably [75]. In this regard, SC-focused interventions could support a more realistic attribution considering the common humanity perspective [5] and may help prevent the self-reinforcing mechanisms of self-doubts and deformed self-reflection of working style that are associated with the impostor profile [2,6,76]. Such interventions

could also be preventive and sooth predictive factors such as social anxiety or socially prescribed perfectionism [51,77]. Remarkably, an intervention for social anxiety could be effective, considering the relationship between isolation and competence doubt in our path-analysis model, as well as associations of isolation with competence doubt, alienation and other-self divergence in the regression models. Furthermore, individuals with IP would benefit from self-efficacy support, which is an expected effect of SC-related interventions [38].

Based on our findings, mindfulness-based and SC-focused interventions may have multiple beneficial effects related to impostor feelings when implemented in university settings [8]. Both mindfulness-based and SC-centered interventions, such as Mindful Self-Compassion or Mindfulness-Based Stress Reduction, generally improve mental health [78]. Their group-based formats are particularly valuable, as they may help reduce students' feelings of isolation and overidentification with their own struggles, which are often accompanied by the belief that "I am the only one facing these issues." Such collective interventions appear especially promising for individuals who share the same social role and identification with a particular university community.

Furthermore, anchoring in the present moment, which is at the core of mindfulness-based interventions, supports the ability to observe one's thoughts from a distance and with clarity, thereby reducing their ruminative potential [79]. However, applying these skills to daily life can be challenging. Therefore, embedding these practices directly within the university environment, which is stressful, especially for students with low SC [80], could be more beneficial than implementing them exclusively in clinical contexts. Moreover, SC-focused interventions in universities may support intrinsic motivation and academic engagement, thereby promoting a well-being-friendly academic environment [8,81].

Finally, students struggling with isolation and overidentification may feel blocked from seeking support and may be imprisoned in their own feelings. Incorporating elements of SC interventions into the university curricula (within courses addressing self-management or well-being at work) can provide students and teachers with accessible tools to counteract impostor feelings and strengthen their psychological resilience [8,82].

## Limitations and further research

Our study had some methodological limitations. The research was based on a cross-sectional design, which does not allow for causal inferences. Under these circumstances, path-analysis methodology can only be used to test for the plausibility of the proposed directional model, including its direct and indirect relationships, rather than to establish causality [83].

Moreover, researchers such as Maxwell et al. [84] argued against using cross-sectional data in mediation analyses, because these may cause biased parameter estimates [84] Our research emphasized the exploratory and associative character of pathways in the path-analysis model, avoiding causal inferences in line with the approach of associational variable analysis [85]. Therefore, future research should validate the relationship between SC and IP using a longitudinal or experimental design.

Additional methodological bias may arise from the non-probabilistic sampling strategy of convenience choice. Further, our results might be influenced by the use of the self-report measures collected within a single survey, which may increase the risk of common method variance and potentially inflate the observed associations between the studied variables.Furthermore, the respondents were Czech university students, which implies a specific context in terms of mental health and well-being [86–88]. The research sample lacked variability because the study programs were similar, and all respondents belonged to one faculty. However, this fact makes our results highly relevant for employers in the economic and business sectors of the Czech Republic because the FEM at the CZU has the highest number of students compared with other economic faculties in the Czech Republic. Furthermore, the selection of this research population aligned with the necessity to study SC training in non-caring professions [34]. However, our study focused more on the initial analysis of the specifics of SC training related to IP.

Additionally, the sample was not balanced according to sex, with 368 women and 232 men. Although this roughly corresponds to the sex structure of FEM students, it may slightly affect the results, considering the sex differences in SC levels [89,90] detected also in the population of university students in the Czech Republic [91]. Overall, our study identified higher isolation, overidentification and lower mindfulness by women, which is consistent with the meta-analytical study that proved slightly lower self-compassion by women [89]. Nevertheless, our results should be validated by future research using a more gender-balanced sample.

Another relevant aspect concerns potential gender differences in the IPP30 subscale competence doubt. Previous research on impostor feelings has consistently shown that women tend to report higher levels of impostorism than men, particularly in domains that emphasize achievement and evaluation [1,21]. In our study, women also demonstrated slightly higher scores on competence doubt, which may reflect a stronger internalization of performance standards and greater susceptibility to self-critical comparison processes. These findings might be partly explained by socialization factors, such as gendered expectations about competence and success or by contextual influences in academic and organizational settings that reinforce self-doubt in women despite comparable objective performance. Generally, the gender differences identified in our study warrant further investigation, particularly regarding the specificity of the SC–IP relationships. Future research should focus on the longitudinal interactional development of these constructs to better understand potential cause-and-effect relationships. Furthermore, an experimental design could help reduce the bias inherent to self-reported questionnaires, such as recall errors, social desirability or reference bias, specifically in college students [92]. Incorporating behavioral measures, observation reports, or experimental manipulation would provide a more comprehensive and objective assessment of SC and IP [93]. Such insights could be especially valuable for the development of interventions, helping determine whether IP or SC functions as the primary cause, and should be prioritized in treatment approaches.

## Author contributions

**Conceptualization:** Kristýna Krejčová, Lucie Kvasničková Stanislavská.

**Data curation:** Kristýna Krejčová, Fabio Ibrahim.

**Formal analysis:** Kristýna Krejčová, Lucie Kvasničková Stanislavská, Fabio Ibrahim.

**Investigation:** Kristýna Krejčová.

**Methodology:** Kristýna Krejčová, Lucie Kvasničková Stanislavská, Fabio Ibrahim.

**Project administration:** Lucie Kvasničková Stanislavská, Fabio Ibrahim.

**Software:** Kristýna Krejčová.

**Supervision:** Lucie Kvasničková Stanislavská.

**Validation:** Kristýna Krejčová, Lucie Kvasničková Stanislavská, Fabio Ibrahim.

**Visualization:** Kristýna Krejčová.

**Writing – original draft:** Kristýna Krejčová.

**Writing – review & editing:** Lucie Kvasničková Stanislavská, Fabio Ibrahim.

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
