## [Decision Letter · Decision Letter 0]

8 Aug 2025

PONE-D-25-34227Self-compassion as a protective factor: Examining its specific role in the impostor phenomenonPLOS ONE

Dear Dr. Krejčová,

Thank you for submitting your manuscript to PLOS ONE. After careful consideration, we feel that it has merit but does not fully meet PLOS ONE’s publication criteria as it currently stands. Therefore, we invite you to submit a revised version of the manuscript that addresses the points raised during the review process.

We look forward to receiving your revised manuscript.

Kind regards,

I Gede Juanamasta

Academic Editor

PLOS ONE

Journal Requirements:

Reviewers' comments:

Reviewer's Responses to Questions

**Comments to the Author**

1. Is the manuscript technically sound, and do the data support the conclusions?

Reviewer #1: Yes

Reviewer #2: Yes

Reviewer #3: No

2. Has the statistical analysis been performed appropriately and rigorously? 

Reviewer #1: Yes

Reviewer #2: Yes

Reviewer #3: No

3. Have the authors made all data underlying the findings in their manuscript fully available?

Reviewer #1: Yes

Reviewer #2: Yes

Reviewer #3: No

4. Is the manuscript presented in an intelligible fashion and written in standard English?

Reviewer #1: No

Reviewer #2: Yes

Reviewer #3: No

5. Review Comments to the Author

Reviewer #1: Overall Summary:

This manuscript explores the relationship between self-compassion (SC) and the impostor phenomenon (IP) among Czech university students. The study uses the Impostor-Profile 30 (IPP30) and the Self-Compassion Scale (SCS) to analyze data from 601 students. The findings suggest that negative dimensions of self-compassion (like isolation and overidentification) are more strongly associated with IP, particularly with competence doubt.

Some Suggestions:

1. Abstract:

- The abstract provides a good overview, but could be more concise. Consider focusing on the most significant findings.

- Specify the type of statistical analysis used (e.g., correlational analysis, regression model, structural equation modelling).

2. Introduction:

- The introduction effectively sets the stage by describing the impostor phenomenon and its relevance, especially in student populations.

- The rationale for focusing on university students could be further expand.

3. Literature Review/Background:

- The literature review covers relevant concepts (impostor phenomenon, self-compassion) and instruments (IPP30, SCS).

- It could benefit from a more critical evaluation of existing literature, highlighting gaps the current study addresses.

- In the section on Self-compassion, clarify how the IP and SC relate to reality.

4. Methods:

- The description of participants, measures, and procedure is thorough.

- The sample size is justified, and the data collection method is appropriate.

- Mention the specific version of IBM SPSS Statistics used.

- Give reasons for using a double translation method from English to Czech. Expand on the forward-backward translation procedure.

5. Results:

- The results section clearly presents the descriptive statistics and correlational analyses.

- The use of tables is effective for summarizing data.

- The SEM results should be reported with more detail (e.g., specific path coefficients, fit indices).

6. Discussion:

- The discussion relates the findings back to the hypotheses and existing literature.

- The discussion could be strengthened by addressing the limitations more explicitly and suggesting directions for future research.

- Expand on the practical implications of the findings for interventions targeting impostorism.

7. Tables and Figures:

- Ensure all tables and figures are properly labeled and referenced in the text.

- Figure 1 (path analysis model) is mentioned but not fully discussed in terms of its implications.

8. References:

- Update references in 2024/2025 and check it is correctly formatted.

- Update the studies’ information and ensure they align with the context they were cited in.

Specific Suggestions:

- Focus on Key Findings: In the abstract and discussion, highlight the most important and novel findings of the study.

- Address Limitations: Acknowledge the limitations of the study, such as the specific population (Czech university students), the cross-sectional design, and the potential for self-report bias.

- Expand on Implications: Discuss the implications of the findings for designing interventions to reduce impostor feelings and promote self-compassion, particularly in university settings.

- Suggest sending the manuscript for English proofreading before acceptance.

Thank you.

Reviewer #2: Thank you for the opportunity to review your manuscript. You provide a compelling case for investigating the IP and SC relationship. The following are some thoughts to help take your work to the next level.

- I noticed that "self-concept", "self-attitude" and "self-image' are used in the manuscript, and in some instances, are used interchangeably - are they referring to the same or different concept(s)? Naive readers may not be familiar with nuances.

- p. 3 line 39 - Why employers suffering from the IP?

- p. 8 line 165 - what is "mutual" correlation?

- p. 8 line 167 - should it be "lower levels of the impostor phenomenon..."?

- p. 8 line 178 - please provide % in addition to n's

- p.9 lines 197-198 - how do the six subscales map onto the core elements provide on p. 4 lines 78-79? To me, they appear quite different, not well aligned.

- p.10 lines 210-213 - the logic between the two sentences is contradictory. Why would one use separate subscale scores in the absence of evidence for clear factor structure as reported in the literature??

- p. 15 line 293-297 - - what theory is it based on? or is it purely based on the initial correlational analyses? That is, why for example, in Figure 1, self-kindness feeds into common humanity, and isolation feeds into overidentification?? The reviewed literature does not offer any justification for the model tested.

- p. 16 line 312 - should it be "....with scores in NEGATIVE subscales slightly prevailing"?

- p. 18 line 347 - what does "treatable" refer to??

Reviewer #3: The present manuscript examines the relations between the Impostor Phenomenon and self-compassion in a sample of 601 students. The findings are mostly in line with expectations.

Overall, I was looking forward to reading the manuscript as the research question is interesting. However, my enthusiasm was dampened upon reading the manuscript. I provide several comments below. Taking my observations into account, the manuscript does not meet the standards for a contribution to the knowledge of the field. Accordingly, I recommend rejecting the manuscript for publication.

(1) Discrepancy between data and language: I am surprised that authors often provide causal interpretations and, thus, over-interpret the data and findings heavily. Considering the cross-sectional nature of the data assumptions about SC being a "protective factor" "influences etc. are impossible and inappropriate.

(2) Introduction and literature review: Overall, the introduction reads very awkwardly and lacks flow that provides readers with a clear rationale of the study and the hypotheses. There are many instances where one-sentence paragraphs are given that appear to be disconnected from the remainder of the manuscript. Also there are several typos and grammatical errors that sometimes even alter the meaning of the words (e.g., "employers" vs "employees").

(3) Description of results: The results are displayed and reported very in transparently. For example, the tables contain all coefficients (e.g., one- and two-tailed tests of significance) and it is up to readers to find and seek the relevant coefficients. Also, there are many inconsistencies that are, again, up to the reader to solve (e.g., statistically significant with negligible effect size). The use of K-S tests for testing is questionable when considering the large sample size and how easily minor deviations from the normal distribution are detected. F-test is reported without degrees of freedom. RMSEA = .41 indicates are very ill-fitting model.

(4) Mediation on basis of cross-sectional data: There is robust evidence that testing indirect effects/mediation analyses on basis of cross-sectional data produce biased parameter estimates (see the works by Maxwell and Cole; e.g., 2007). Hence, findings from this line of research can and should not be trusted.

Maxwell, S. E., & Cole, D. A. (2007). Bias in cross-sectional analyses of longitudinal mediation. Psychological methods, 12(1), 23–44. https://doi.org/10.1037/1082-989X.12.1.23

(5) Novelty: Even when taking the issues noted above aside, it is unclear what the novel contribution of this study is to the field.

(6) While I applaud authors' efforts to provide a link to the open data, it does not work and leads nowhere.

6. PLOS authors have the option to publish the peer review history of their article (what does this mean?). If published, this will include your full peer review and any attached files.

Reviewer #1: No

Reviewer #2: No

Reviewer #3: No

---

## [Author Response · Author response to Decision Letter 1]

2 Sep 2025

Reviewer #1: Overall Summary:

This manuscript explores the relationship between self-compassion (SC) and the impostor phenomenon (IP) among Czech university students. The study uses the Impostor-Profile 30 (IPP30) and the Self-Compassion Scale (SCS) to analyze data from 601 students. The findings suggest that negative dimensions of self-compassion (like isolation and overidentification) are more strongly associated with IP, particularly with competence doubt.

Some Suggestions:

1. Abstract:

- The abstract provides a good overview, but could be more concise. Consider focusing on the most significant findings.

- Specify the type of statistical analysis used (e.g., correlational analysis, regression model, structural equation modelling).

Authors’ response:

In accordance with the feedback, we revised the abstract to improve clarity and methodological transparency. We streamlined several sentences for clarity while retaining their original meaning; explicitly stated the types of analyses applied and clarified the SEM results.

2. Introduction:

- The introduction effectively sets the stage by describing the impostor phenomenon and its relevance, especially in student populations.

- The rationale for focusing on university students could be further expand.

Authors’ response:

We have expanded the rationale for focusing on university students by adding practical relevance and empirical support for our decision.

3. Literature Review/Background:

- The literature review covers relevant concepts (impostor phenomenon, self-compassion) and instruments (IPP30, SCS).

- It could benefit from a more critical evaluation of existing literature, highlighting gaps the current study addresses.

Authors’ response:

We have added a critical evaluation in the general description of the IP syndrome, highlighting the predominance of research on medical students. Further, in the section “IP and SC: well-being-related contexts and differences,” we emphasised the lack of studies examining the mutual relationships between these two concepts, which was already mentioned in the introduction.

- In the section on Self-compassion, clarify how the IP and SC relate to reality.

Authors’ response:

The description of relationships of the SC and IP to reality was added in the closing paragraph of the Self-compassion section.

4. Methods:

- The description of participants, measures, and procedure is thorough.

- The sample size is justified, and the data collection method is appropriate.

- Mention the specific version of IBM SPSS Statistics used.

Authors’ response:

The version of IBM SPSS Statistics was assigned more specifically.

- Give reasons for using a double translation method from English to Czech. Expand on the forward-backward translation procedure.

Authors’ response:

We are grateful to the reviewer for pointing out the improper description of the translation process. The information was corrected and specified.

5. Results:

- The results section clearly presents the descriptive statistics and correlational analyses.

- The use of tables is effective for summarizing data.

- The SEM results should be reported with more detail (e.g., specific path coefficients, fit indices).

Authors’ response:

We extended the interpretation of fit indices and added a specific description of the path coefficients.

6. Discussion:

- The discussion relates the findings back to the hypotheses and existing literature.

- The discussion could be strengthened by addressing the limitations more explicitly and suggesting directions for future research.

Authors’ response:

We have expanded the section on study limitations to include more specific details regarding the sample’s nationality, the reliance on self-report measures, and the cross-sectional design. Based on these additions, we also elaborated on potential directions for future research, highlighting the need for longitudinal and experimental approaches as well as alternative assessment methods (e.g., behavioral measures, informant reports). To strengthen this section, we incorporated recent scientific articles that further support our discussion of limitations and methodological alternatives.

- Expand on the practical implications of the findings for interventions targeting impostorism.

Authors’ response:

We thank the reviewer for this important suggestion. The section on practical implications has been extended in light of our findings and their discussion. Specifically, we now elaborate on the advantages of implementing self-compassion–centered interventions in group settings, which may help reduce students’ feelings of isolation and overidentification. We also highlight that embedding such interventions directly in the university environment could facilitate the transfer of self-compassionate responses into students’ daily academic lives, which are often shaped by high achievement orientation. Furthermore, we discuss the potential positive impact on students’ academic engagement and performance, as well as the broader benefits of incorporating elements of self-compassion–centered and mindfulness-based interventions into the university curriculum, particularly within courses addressing self-management and well-being at work.

7. Tables and Figures:

- Ensure all tables and figures are properly labeled and referenced in the text.

- Figure 1 (path analysis model) is mentioned but not fully discussed in terms of its implications.

Authors’ response:

The description of the SEM model was extended in the Results section, and further information on the practical implications was added in the Theoretical and Practical Implications section.

8. References:

- Update references in 2024/2025 and check it is correctly formatted.

- Update the studies’ information and ensure they align with the context they were cited in.

Authors’ response:

The references were updated and corrected in formatting where necessary.

Specific Suggestions:

- Focus on Key Findings: In the abstract and discussion, highlight the most important and novel findings of the study.

- Address Limitations: Acknowledge the limitations of the study, such as the specific population (Czech university students), the cross-sectional design, and the potential for self-report bias.

- Expand on Implications: Discuss the implications of the findings for designing interventions to reduce impostor feelings and promote self-compassion, particularly in university settings.

- Suggest sending the manuscript for English proofreading before acceptance.

Authors’ response:

The responses to the specific suggestions were mostly already addressed in the previous text. Furthermore, we appreciate the recommendation to have the manuscript proofread in English, which we have now done using the Editage proofreading service.

Reviewer #2: Thank you for the opportunity to review your manuscript. You provide a compelling case for investigating the IP and SC relationship. The following are some thoughts to help take your work to the next level.

- I noticed that "self-concept", "self-attitude" and "self-image' are used in the manuscript, and in some instances, are used interchangeably - are they referring to the same or different concept(s)? Naive readers may not be familiar with nuances

Authors’ response:

We thank the reviewer for this important note, and we acknowledge that readers might be confused by this kind of inconsistency. Therefore, we have decided to replace both self-image and self-concept with self-attitude, a broader term encompassing cognitive, emotional, and motivational dimensions (as inherent in the concept of attitude itself).

- p. 3 line 39 - Why employers suffering from the IP?

- p. 8 line 165 - what is "mutual" correlation?

- p. 8 line 167 - should it be "lower levels of the impostor phenomenon..."?

- p. 8 line 178 - please provide % in addition to n's

Authors’ response:

We corrected/removed all mistakes and redundancies and added % where required.

- p.9 lines 197-198 - how do the six subscales map onto the core elements provide on p. 4 lines 78-79? To me, they appear quite different, not well aligned.

Authors’ response:

The subscales do not match the core elements of the IP. For better clarity, we described all scales in the Research background section.

- p.10 lines 210-213 - the logic between the two sentences is contradictory. Why would one use separate subscale scores in the absence of evidence for clear factor structure as reported in the literature??

Authors’ response:

The use of separate subscale scores of the Self-Compassion Scale (SCS) is in line with the recommendations of the scale’s author (Neff, 2003; Neff, 2016), who explicitly allows the reporting of both total and subscale scores. Moreover, numerous empirical studies have analysed the subscales individually to capture the multidimensional nature of self-compassion. While some studies support a higher-order factor structure, others have highlighted methodological concerns regarding the validity of the total score (e.g., López et al., 2015; Muris et al., 2022). These authors recommend reporting partial scores rather than relying solely on the total score. Nevertheless, we thank the reviewer for the insightful comments that inspired us to clarify this in the Measures section.

- p. 15 line 293-297 - - what theory is it based on? or is it purely based on the initial correlational analyses? That is, why for example, in Figure 1, self-kindness feeds into common humanity, and isolation feeds into overidentification?? The reviewed literature does not offer any justification for the model tested.

Authors’ response:

Yes, the model is partially based on correlation analyses but is also grounded in theoretical considerations. For instance, Dreisoerner et al. (2020) note that “self-kindness may be linked to common humanity and isolation” (p. 25). They describe a vicious circle of self-criticism leading to shame, reduced socially oriented behaviour, and ultimately social exclusion. Furthermore, the connection between Isolation and Over-identification in our model is supported by Cleare et al. (2018), who reported the highest intercorrelation between these two subscales within the six-factor correlated model of the SCS. In addition to this theoretical foundation, our final model was selected after testing alternative models with poorer fit. We have now clarified this by adding information about the relevant literature both in the section on theoretical background and in the analytic plan.

- p. 16 line 312 - should it be "....with scores in NEGATIVE subscales slightly prevailing"?

Authors’ response:

We thank the reviewer for the important recommendation. This information is correct, but there was a typo in the Result section; we corrected it to ensure clarity and consistency.

- p. 18 line 347 - what does "treatable" refer to??

Authors’ response:

It was connected to the self-compassion level; we reformulated the text and replaced the improper verb with a better synonym.

Reviewer #3: The present manuscript examines the relations between the Impostor Phenomenon and self-compassion in a sample of 601 students. The findings are mostly in line with expectations.

Overall, I was looking forward to reading the manuscript as the research question is interesting. However, my enthusiasm was dampened upon reading the manuscript. I provide several comments below. Taking my observations into account, the manuscript does not meet the standards for a contribution to the knowledge of the field. Accordingly, I recommend rejecting the manuscript for publication.

(1) Discrepancy between data and language: I am surprised that authors often provide causal interpretations and, thus, over-interpret the data and findings heavily. Considering the cross-sectional nature of the data assumptions about SC being a "protective factor" "influences etc. are impossible and inappropriate.

Authors’ response:

We thank the reviewer for this valuable comment. We would like to clarify that our study never intended to make causal claims. Our theoretical framework and analytic approach were designed to examine associations, not causal mechanisms.

We acknowledge, however, that some of our original wording could have unintentionally suggested causal relationships. In response to the reviewer’s concern, we systematically revised the manuscript — including the title, abstract, introduction, results, discussion, and implications sections — to ensure that the language consistently reflects correlational, non-causal interpretations.

While the scientific claims of our study remain unchanged, we believe the revised wording now removes any risk of causal overinterpretation.

(2) Introduction and literature review: Overall, the introduction reads very awkwardly and lacks flow that provides readers with a clear rationale of the study and the hypotheses. There are many instances where one-sentence paragraphs are given that appear to be disconnected from the remainder of the manuscript. Also there are several typos and grammatical errors that sometimes even alter the meaning of the words (e.g., "employers" vs "employees").

Authors’ response:

We thank the reviewer for the feedback, which helped us identify and correct typographical and grammatical errors. We have also revised the structure of the Introduction and Research Background sections to improve coherence and provide a clearer rationale for the study. However, please note that the track-changes mode did not allow us to fully display certain revisions, especially those involving the restructuring and merging of paragraphs. These changes have nevertheless been carefully implemented in the manuscript.

(3) Description of results: The results are displayed and reported very in transparently. For example, the tables contain all coefficients (e.g., one- and two-tailed tests of significance) and it is up to readers to find and seek the relevant coefficients. Also, there are many inconsistencies that are, again, up to the reader to solve (e.g., statistically significant with negligible effect size). The use of K-S tests for testing is questionable when considering the large sample size and how easily minor deviations from the normal distribution are detected. F-test is reported without degrees of freedom. RMSEA = .41 indicates are very ill-fitting model.

Authors’ response:

Our original intention was to provide full transparency by reporting all coefficients from the statistical output. However, we acknowledge that this approach may have appeared overloaded and less reader-friendly. In response, we undertook a systematic revision of our results reporting to make it clearer and more consistent with APA guidelines. Further, we replaced the Kolmogorov–Smirnov test with the Shapiro–Wilk test, which is generally preferred for testing normality, and clarified in the manuscript that minor deviations were expected given the large sample size, but do not compromise the robustness of parametric analyses. We also added the degree of freedom in the F-test interpretation and included also the confidence intervals to be more consistent with recommendations of APA (7th edition).

We sincerely thank the reviewer for drawing attention to the value of RMSEA, which was mistakenly reported as 0.41 in the text due to a typographical. We would like to clarify that the correct value is RMSEA = 0.041, as was accurately presented in the table 7. We have revised the text accordingly to ensure correctness and consistency.

(4) Mediation on basis of cross-sectional data: There is robust evidence that testing indirect effects/mediation analyses on basis of cross-sectional data produce biased parameter estimates (see the works by Maxwell and Cole; e.g., 2007). Hence, findings from this line of research can and should not be trusted.

Maxwell, S. E., & Cole, D. A. (2007). Bias in cross-sectional analyses of longitudinal mediation. Psychological methods, 12(1), 23–44. https://doi.org/10.1037/1082-989X.12.1.23

Authors’ response:

We sincerely thank the reviewer for this important note. We fully agree that mediation analyses assume temporal ordering, which cannot be established with cross-sectional data. In line with Maxwell et al. (2011), we acknowledge that parameter estimates

---

## [Decision Letter · Decision Letter 1]

6 Oct 2025

PONE-D-25-34227R1Self-compassion and the impostor phenomenon: Associations and implicationsPLOS ONE

Dear Dr. Krejčová,

Thank you for submitting your manuscript to PLOS ONE. After careful consideration, we feel that it has merit but does not fully meet PLOS ONE’s publication criteria as it currently stands. Therefore, we invite you to submit a revised version of the manuscript that addresses the points raised during the review process.

We look forward to receiving your revised manuscript.

Kind regards,

I Gede Juanamasta

Academic Editor

PLOS ONE

Journal Requirements:

Reviewers' comments:

Reviewer's Responses to Questions

**Comments to the Author**

1. If the authors have adequately addressed your comments raised in a previous round of review and you feel that this manuscript is now acceptable for publication, you may indicate that here to bypass the “Comments to the Author” section, enter your conflict of interest statement in the “Confidential to Editor” section, and submit your "Accept" recommendation.

Reviewer #3: All comments have been addressed

2. Is the manuscript technically sound, and do the data support the conclusions?

Reviewer #3: Partly

3. Has the statistical analysis been performed appropriately and rigorously? 

Reviewer #3: No

4. Have the authors made all data underlying the findings in their manuscript fully available?

Reviewer #3: Yes

5. Is the manuscript presented in an intelligible fashion and written in standard English?

Reviewer #3: Yes

6. Review Comments to the Author

Reviewer #3: I thank the authors for addressing my comments in a careful revision of the manuscript. I have read the authors' responses and the manuscript to the comments closely and found that the manuscript was strengthened. However, some points require attention that should be addressed in a second revision of the manuscript:

(1) Authors note in the manuscript that the CIPS is unidimensional, which is not in line with the state-of-the-art of the literature, as the CIPS has been found to be presented by three dimensions in addition to a general factor (Brauer & Proyer, 2025). Hence, the claim on the "unidimensional scale method" when referring to the CIPS requires revision.

Brauer, K., & Proyer, R. T. (2025). Understanding the Clance Impostor Phenomenon Scale through the lens of a bifactor model. European Journal of Psychological Assessment, 41(2), 108–116. https://doi.org/10.1027/1015-5759/a000786

(2) The IP is not a clinical construct. Hence, speaking of "symptomatology" is misleading and requires revision.

(3) Tables should be formatted in line with APA recommendations instead of a copy-paste of the SPSS output.

(4) Asterisks *and* p-values in Table 5 are redundant. Please report either.

(5) As authors note, the sample size is large, which relates to oversensitive tests of normal distribution (S-W test). Please report and interpret normality on basis of skewness and kurtosis parameters, as the S-W test is oversensitive to negligible deviations from the normal distribution. If those parameters are < 1, I suggest switching the correlation analyses to Pearson, as you do not analyze categorical data but continuous data in the sense of mean scores.

(6) I have missed analyses regarding demographic variables such as age and gender.

(7) Change "SEM" to "path analysis" since the figure of the model indicates that only manifest variables were considered.

(8) Clarify the estimator used in the path analysis. MLR is recommended.

(9) Please provide syntaxes for the analyses. Currently, only a data file is provided.

7. PLOS authors have the option to publish the peer review history of their article (what does this mean?). If published, this will include your full peer review and any attached files.

Reviewer #3: No

---

## [Author Response · Author response to Decision Letter 2]

15 Oct 2025

Response to Reviewers

Reviewer #3: I thank the authors for addressing my comments in a careful revision of the manuscript. I have read the authors' responses and the manuscript to the comments closely and found that the manuscript was strengthened. However, some points require attention that should be addressed in a second revision of the manuscript:

(1) Authors note in the manuscript that the CIPS is unidimensional, which is not in line with the state-of-the-art of the literature, as the CIPS has been found to be presented by three dimensions in addition to a general factor (Brauer & Proyer, 2025). Hence, the claim on the "unidimensional scale method" when referring to the CIPS requires revision.

Brauer, K., & Proyer, R. T. (2025). Understanding the Clance Impostor Phenomenon Scale through the lens of a bifactor model. European Journal of Psychological Assessment, 41(2), 108–116. https://doi.org/10.1027/1015-5759/a000786

Authors’ response:

We are grateful to the reviewer for drawing our attention to this recent and insightful study. Although we still have a rationale for using the IPP-30 instead of the CIPS in our research, we have gladly incorporated the cited paper into the manuscript to enhance the accuracy and clarity of our argumentation.

(2) The IP is not a clinical construct. Hence, speaking of "symptomatology" is misleading and requires revision.

Authors’ response:

We agree with the reviewer that the term “symptomatology” is somewhat inappropriate given the nature of the IP. In line with the study cited in the corresponding paragraph, we have replaced it with the term “syndrome.”

(3) Tables should be formatted in line with APA recommendations instead of a copy-paste of the SPSS output.

Authors’ response:

We are grateful to the reviewer for this helpful remark that has helped us to improve the graphical presentation and readability of the manuscript.

In response, we have reformatted all tables in accordance with APA recommendations.

(4) Asterisks *and* p-values in Table 5 are redundant. Please report either.

Authors’ response:

Following the reviewer’s suggestion, we revised Table 5 (now Table 7) by removing the redundant “Sig.” rows and keeping only the significance asterisks, which aligns the table formatting with APA guidelines.

(5) As authors note, the sample size is large, which relates to oversensitive tests of normal distribution (S-W test). Please report and interpret normality on basis of skewness and kurtosis parameters, as the S-W test is oversensitive to negligible deviations from the normal distribution. If those parameters are < 1, I suggest switching the correlation analyses to Pearson, as you do not analyze categorical data but continuous data in the sense of mean scores.

Authors’ response:

We thank the reviewer for this valuable suggestion. In line with the recommendation, we assessed normality using skewness and kurtosis parameters, which indicated only minor deviations from normality. Given the large sample size and the approximately normal distribution of the data, we conducted the correlation analyses using Pearson’s correlation coefficient, which was considered appropriate for the continuous scores used in this study.

(6) I have missed analyses regarding demographic variables such as age and gender.

Authors’ response:

We thank the reviewer for this helpful comment. In response, we conducted additional analyses to examine the potential effects of gender using independent-samples t-tests. The results indicated that gender differences were mostly non-significant, with females scoring higher on isolation, overidentification, and competence doubt, and males scoring higher on mindfulness. These differences were also discussed in the Discussion section. The analysis of age effects was not performed, as the age distribution in our sample was narrow and homogeneous, which would not allow for meaningful comparisons.

(7) Change "SEM" to "path analysis" since the figure of the model indicates that only manifest variables were considered.

Authors’ response:

We have changed the description of the methodological basis of our model from SEM to path analysis to ensure consistency with the nature of the variables represented in the model.

(8) Clarify the estimator used in the path analysis. MLR is recommended.

Authors’ response:

We thank the reviewer for this comment. The path analysis was conducted in IBM SPSS Amos using the Maximum Likelihood (ML) estimator, which is the default estimation method in this software. As indicated in Table 4, only minor deviations from normality were observed. Given the large sample size and approximately normal distribution of the data, the use of ML estimation was considered appropriate.

(9) Please provide syntaxes for the analyses. Currently, only a data file is provided.

Authors’ response:

In response to the reviewer’s comment, we have attached all relevant syntax files used for the analyses conducted in SPSS and Amos. These files are now provided in the Mendeley Data Repository to ensure full transparency and replicability of our analytical procedures.

---

## [Decision Letter · Decision Letter 2]

17 Nov 2025

PONE-D-25-34227R2Self-compassion and the impostor phenomenon: Associations and implicationsPLOS ONE

Dear Dr. Krejčová,

Thank you for submitting your manuscript to PLOS ONE. After careful consideration, we feel that it has merit but does not fully meet PLOS ONE’s publication criteria as it currently stands. Therefore, we invite you to submit a revised version of the manuscript that addresses the points raised during the review process.

We look forward to receiving your revised manuscript.

Kind regards,

I Gede Juanamasta

Academic Editor

PLOS ONE

Journal Requirements:

Reviewers' comments:

Reviewer's Responses to Questions

**Comments to the Author**

1. If the authors have adequately addressed your comments raised in a previous round of review and you feel that this manuscript is now acceptable for publication, you may indicate that here to bypass the “Comments to the Author” section, enter your conflict of interest statement in the “Confidential to Editor” section, and submit your "Accept" recommendation.

Reviewer #3: (No Response)

Reviewer #4: (No Response)

2. Is the manuscript technically sound, and do the data support the conclusions?

Reviewer #3: Partly

Reviewer #4: Partly

3. Has the statistical analysis been performed appropriately and rigorously? 

Reviewer #3: N/A

Reviewer #4: No

4. Have the authors made all data underlying the findings in their manuscript fully available?

Reviewer #3: Yes

Reviewer #4: Yes

5. Is the manuscript presented in an intelligible fashion and written in standard English?

Reviewer #3: Yes

Reviewer #4: Yes

6. Review Comments to the Author

Reviewer #3: I thank the authors for again addressing the open points and comments. Two final points remain that I kindly ask the authors to address.

(1) In response to my comment on refraining from the description of "symptoms" authors note: "We agree with the reviewer that the term “symptomatology” is somewhat inappropriate

given the nature of the IP. In line with the study cited in the corresponding paragraph,

we have replaced it with the term “syndrome.”"

However, considering that a syndrome is defined as "a group of symptoms which consistently occur together" the revision from symptoms to syndrome is inappropriate. Please just rename it "Impostor Phenomenon" in line with the standard literature and the remainder of this manuscript.

(2) Since you report gender, please change "males" to "men" and "females" to "women" in line with APA recommendations on reporting gender.

Reviewer #4: This is my first time reviewing this manuscript, so I should begin by noting that the comments from the previous reviewer were only partially addressed. The authors clarified the use of the IPP-30, adjusted the language related to the impostor syndrome, removed the asterisks and p-values appearing in the same table, interpreted normality using skewness and kurtosis parameters, replaced the term SEM analysis with path analysis, and clarified the estimator used, although they did not apply MLR. They also made the syntax available.

However, the tables could be merged and presented in a clearer, more unified format. In addition, the analyses based on age and gender were only partially completed; these variables should be included as control variables in the path analysis. I also have doubts about the use of paired t-tests to compare the positive and negative SC subscales. Since this is a cross-sectional study, I do not find a valid reason for using this type of analysis.

7. PLOS authors have the option to publish the peer review history of their article (what does this mean?). If published, this will include your full peer review and any attached files.

Reviewer #3: No

Reviewer #4: No

---

## [Author Response · Author response to Decision Letter 3]

21 Nov 2025

Response to Reviewers

Reviewer #3:

I thank the authors for again addressing the open points and comments. Two final points remain that I kindly ask the authors to address.

(1) In response to my comment on refraining from the description of "symptoms" authors note: "We agree with the reviewer that the term “symptomatology” is somewhat inappropriate

given the nature of the IP. In line with the study cited in the corresponding paragraph,

we have replaced it with the term “syndrome.”"

However, considering that a syndrome is defined as "a group of symptoms which consistently occur together" the revision from symptoms to syndrome is inappropriate. Please just rename it "Impostor Phenomenon" in line with the standard literature and the remainder of this manuscript.

Authors’ response:

We thank the reviewer for this comment. While the term “syndrome” does occasionally appear in the literature on the Impostor Phenomenon, we agree with the reviewer that its use is not appropriate in this context, because it implies the presence of a consistent grouping of symptoms, which does not align with the non-clinical nature of the construct. In line with the reviewer’s recommendation and the prevailing terminology in the field, we have therefore replaced the term with “Impostor Phenomenon” throughout the manuscript.

(2) Since you report gender, please change "males" to "men" and "females" to "women" in line with APA recommendations on reporting gender.

Authors’ response:

In accordance with APA recommendations on reporting gender, we have replaced the terms “males” and “females” with “men” and “women” throughout the manuscript. We appreciate the reviewer’s guidance in helping us improve the clarity and inclusivity of our reporting.

Reviewer #4: This is my first time reviewing this manuscript, so I should begin by noting that the comments from the previous reviewer were only partially addressed. The authors clarified the use of the IPP-30, adjusted the language related to the impostor syndrome, removed the asterisks and p-values appearing in the same table, interpreted normality using skewness and kurtosis parameters, replaced the term SEM analysis with path analysis, and clarified the estimator used, although they did not apply MLR. They also made the syntax available.

1/ However, the tables could be merged and presented in a clearer, more unified format.

Authors’ response:

Based on the recommendation, we carefully revised the presentation of all tables to ensure clarity, consistency, and compliance with APA formatting standards.

All tables were reformatted to adopt the same graphical style as Table 1, including consistent typography, alignment, and use of horizontal rules.

In the Table 1, abbreviations of both questionnaires were deleted fot a better consistency with the other tables.

The table reporting gender differences in descriptive statistics was reorganized to improve readability.

Instead of listing gender categories in rows, the table was redesigned so that gender is placed in columns (with separate columns for N, Mean, and SD for men and women).

This format is more compact, reduces unnecessary vertical space

The independent samples test table was substantially simplified.

In accordance with APA recommendations, we removed duplicated lines stemming from the “equal variances assumed / not assumed” output.

For each variable, only the appropriate t-test result (based on Levene’s test) is now reported.

A clear note was added below the table:

“Note. For each variable, Levene’s test was used to determine the appropriate t-test; only the relevant row is reported.”

We believe that these improvements significantly enhance the readability, coherence, and overall quality of the manuscript’s results section.

2/ In addition, the analyses based on age and gender were only partially completed; these variables should be included as control variables in the path analysis.

Authors’ response:

In response, we revised our analytical strategy as follows:

Gender was included as a covariate in the path analysis to evaluate the stability of the associations among latent constructs across demographic subgroups. The revised model demonstrated that the effects of gender were minimal, with standardized regression coefficients ranging from β = –.04 to .13. These values do not reach practical significance and did not meaningfully alter the structure or strength of the relations in the model.

Age was not included in the final model, and we would like to briefly clarify this decision.

Our sample exhibits extremely low age variability (M = 23.46, SD = 3.05). The participants were university students within a very narrow age range.

We believe that including age as a predictor in our path analysis does not improve model validity.

The inclusion of gender did not change any substantive conclusions. The model remained stable, and all associations among the constructs persisted with virtually identical effect sizes. This confirms the robustness of the findings across gender influences.

3/ I also have doubts about the use of paired t-tests to compare the positive and negative SC subscales. Since this is a cross-sectional study, I do not find a valid reason for using this type of analysis.

Authors’ response:

We agree with the reviewer’s viewpoint that, given the cross-sectional nature of our study, the use of paired t-tests to compare the positive and negative self-compassion subscales is not methodologically well-justified. Since this analysis is not essential for addressing our research questions, we have removed it from the manuscript.

We appreciate the reviewer’s careful consideration, which has strengthened the methodological rigour of our work.

---

## [Decision Letter · Decision Letter 3]

18 Dec 2025

PONE-D-25-34227R3

Self-compassion and the impostor phenomenon: Associations and implications

PLOS One

Dear Dr. Krejčová,

Thank you for submitting your manuscript to PLOS ONE. After careful consideration, we have decided that your manuscript does not meet our criteria for publication and must therefore be rejected.

I am sorry that we cannot be more positive on this occasion, but hope that you appreciate the reasons for this decision.

Kind regards,

I Gede Juanamasta

Academic Editor

PLOS One

Reviewers' comments:

Reviewer's Responses to Questions

**Comments to the Author**

1. If the authors have adequately addressed your comments raised in a previous round of review and you feel that this manuscript is now acceptable for publication, you may indicate that here to bypass the “Comments to the Author” section, enter your conflict of interest statement in the “Confidential to Editor” section, and submit your "Accept" recommendation.

Reviewer #3: (No Response)

2. Is the manuscript technically sound, and do the data support the conclusions?

Reviewer #3: No

3. Has the statistical analysis been performed appropriately and rigorously? 

Reviewer #3: No

4. Have the authors made all data underlying the findings in their manuscript fully available?

Reviewer #3: Yes

5. Is the manuscript presented in an intelligible fashion and written in standard English?

Reviewer #3: No

6. Review Comments to the Author

Reviewer #3: The present manuscript is the fourth version of a manuscript describing the study between the IP and self-compassion. Please my comments on the current version of the manuscript below:

- I am grateful to authors that they deleted instances of symptoms and syndrome to acknowledge that the IP is not a categorical variable of clinical entity. I have just discovered the sentence "The selection of this research population aligns with studies that reflect the high incidence of IP..." on p. 4 and "Several studies have focused on the incidence of IP among medical students". Since incidence is defined as "the occurrence, rate, or frequency of a disease, crime, or other undesirable thing" this should be revised accordingly.

Similarly, authors note "Later research pointed out the prevalence of IP regardless of gender, ethnicity, or age." Since prevalence describes the occurrence of clinical entities such as diseases, revisions are required. As noted, the IP is neither a categorical variable nor a clinical entity.

- Tables 2 and 3: The columns N(M) and N(W) provide redundant information and can be deleted.

- Table 4: The row df is redundant.

- Table 5: Half of the coefficients are redundant with the other remaining half

- Table 6: Lower and upper bounds of the confidence interval regarding SK (32.89, 54.35) do neither correspond to the b or beta coefficient (-0.28, -.07). The same is true for OI, M and other variables. This again decreases trust in the findings.

- Table 7 is redundant with the coefficients given in-text. The table can be deleted.

- The current version of the manuscript contains 108 (!) references, which is excessive given that the study is a simple cross-sectional study investigating the associations between two variables.

Considering that the fourth version of this manuscript still contains so many issues and inconsistencies from a theoretical and empirical perspective (e.g., errors in tables), my impression is that the time has come to bring the review process to an end and I recommend rejecting the manuscript for publication.

7. PLOS authors have the option to publish the peer review history of their article (what does this mean?). If published, this will include your full peer review and any attached files.

Reviewer #3: No

- - - - -

---

## [Author Response · Author response to Decision Letter 4]

2 Jan 2026

Response to Reviewers

We sincerely thank Reviewer 3 for the careful and constructive comments, which prompted us to thoroughly re-examine the manuscript. As a result, we identified and corrected several issues that had arisen during the editing of earlier manuscript versions. Below, we respond to the reviewer’s specific comments.

Reviewer #3:

(1) I am grateful to authors that they deleted instances of symptoms and syndrome to acknowledge that the IP is not a categorical variable of clinical entity. I have just discovered the sentence "The selection of this research population aligns with studies that reflect the high incidence of IP..." on p. 4 and "Several studies have focused on the incidence of IP among medical students". Since incidence is defined as "the occurrence, rate, or frequency of a disease, crime, or other undesirable thing" this should be revised accordingly.

Authors’ response:

We thank the reviewer for this comment. The term “incidence” has been replaced with the less clinically loaded term “occurrence.”

(2) Similarly, authors note "Later research pointed out the prevalence of IP regardless of gender, ethnicity, or age." Since prevalence describes the occurrence of clinical entities such as diseases, revisions are required. As noted, the IP is neither a categorical variable nor a clinical entity.

Authors’ response:

The sentence was revised in line with the reviewer’s comment. Although similar terminology has been used in previous literature (e.g., “Prevalence, Predictors, and Treatment of Impostor Syndrome: a Systematic Review“ (doi: 10.1007/s11606-019-05364-1); „Global prevalence of imposter syndrome in health service providers: a systematic review and meta-analysis“ (doi: 10.1186/s40359-025-02898-4), we agree that a non-clinical formulation is more appropriate in the present context.

(3) Tables 2 and 3: The columns N(M) and N(W) provide redundant information and can be deleted.

Authors’ response:

Tables 2 and 3 have been revised in accordance with the reviewer’s comments. We used standardised statistical output, but we agree that the columns N(M) and N(W) were redundant and have therefore been removed.

4/ Table 4: The row df is redundant.

Authors’ response:

Column df was removed from Table 4.

5/ Table 5: Half of the coefficients are redundant with the other remaining half.

Authors’ response:

Table 5 was originally presented in a symmetrical format, which is standard for correlation matrices and was chosen for clarity. However, the table has been revised in response to the reviewer’s comment.

6/ Lower and upper bounds of the confidence interval regarding SK (32.89, 54.35) do neither correspond to the b or beta coefficient (-0.28, -.07). The same is true for OI, M and other variables. This again decreases trust in the findings.

Authors’ response:

Thank you for noting this inconsistency. This was a formatting error that occurred during the editing of the table. Specifically, the confidence interval values were inadvertently shifted by one row due to the inclusion of the constant row from the SPSS output. The underlying statistical results were correct, but the values were misaligned in the table. The table has now been corrected, and all confidence intervals correspond appropriately to the reported regression coefficients.

7/ Table 7 is redundant with the coefficients given in-text. The table can be deleted.

Authors’ response:

We consider Table 7 a regular summary of model fitting; however, we have removed it in response to the reviewer’s comment.

8/ The current version of the manuscript contains 108 (!) references, which is excessive given that the study is a simple cross-sectional study investigating the associations between two variables.

Authors’ response:

We provided the literature review and discussion we considered appropriate to the manuscript. Some references were also changed during the review process on the request of reviewers – the first version submitted in PLOS ONE had 78 references. The number of references reflects the need to adequately cover two theoretically and empirically distinct constructs—the impostor phenomenon and self-compassion—both of which have extensive and partly independent bodies of literature. Nevertheless, we are willing to reduce the reference list if necessary.

9/ Reviewer also states “No” to the “5. Is the manuscript presented in an intelligible fashion and written in standard English?“.

No specific concerns regarding language clarity or correctness were provided. We submitted a language editing certificate, and the text has not been changed in a manner that would justify a different evaluation between review rounds.

In the second round of the revisions (6. 10. 2025), the same reviewer responded to the same question positively:

“5. Is the manuscript presented in an intelligible fashion and written in standard English?

Reviewer #3: Yes“

---

## [Decision Letter · Decision Letter 4]

5 Feb 2026

PONE-D-25-34227R4Self-compassion and the impostor phenomenon: Associations and implicationsPLOS One

Dear Dr. Krejčová,

Thank you for submitting your manuscript to PLOS ONE. After careful consideration, we feel that it has merit but does not fully meet PLOS ONE’s publication criteria as it currently stands. Therefore, we invite you to submit a revised version of the manuscript that addresses the points raised during the review process.

I agree with the reviewers about the need for further revisions of the manuscript

We look forward to receiving your revised manuscript.

Kind regards,

Diego A. Forero, MD; PhD

Academic Editor

PLOS One

Journal Requirements:

Additional Editor Comments (if provided):

Reviewers' comments:

Reviewer's Responses to Questions

**Comments to the Author**

1. If the authors have adequately addressed your comments raised in a previous round of review and you feel that this manuscript is now acceptable for publication, you may indicate that here to bypass the “Comments to the Author” section, enter your conflict of interest statement in the “Confidential to Editor” section, and submit your "Accept" recommendation.

Reviewer #5: (No Response)

Reviewer #6: All comments have been addressed

2. Is the manuscript technically sound, and do the data support the conclusions?

Reviewer #5: Partly

Reviewer #6: Yes

3. Has the statistical analysis been performed appropriately and rigorously? 

Reviewer #5: Yes

Reviewer #6: Yes

4. Have the authors made all data underlying the findings in their manuscript fully available?

Reviewer #5: No

Reviewer #6: Yes

5. Is the manuscript presented in an intelligible fashion and written in standard English?

Reviewer #5: Yes

Reviewer #6: Yes

6. Review Comments to the Author

Reviewer #5: Conceptual contribution

Clarify what the IPP30 adds conceptually (vs clance or other measures), not just psychometrically.

Detail why ‘competence doubt’ is the critical IP dimension linking to isolation and overidentification.

Explicitly state how this refines or challenges existing models

Data analysis

Path model: implies mechanism and directionality.

Although the authors appropriately note that directional paths are not causal, the interpretation of indirect pathways and the narrative sequencing of variables implies a process model that cannot be supported with cross-sectional data.

Several paths are plausibly reversible or reciprocal, particularly the links between isolation and overidentification, competence doubt and working style, and competence doubt and other-self divergence. The cited literature supports these variables’ covariance but does not establish temporal ordering and the unidirectional paths are not sufficiently justified vs other plausible directionality that are not tested.

I recommend testing and reporting fit comparisons for at least one plausible alternative directional model for each key pathway, and/or providing a stronger justification for why the chosen ordering is privileged over alternatives

As currently presented, the directional arrows risk implying mechanistic interpretation that cannot be supported by the present design.

The choice to only regress competence doubt feels post hoc- justify why the choice was made not to regress other facets of IP. Not including the rest of the IP subscales undermines the use of the IPP30. This is a key to what this study adds to the literature.

Gender was analyzed and reported but not well interpreted. This factor was also not included in hypotheses. It felt like an important element in the results but then dropped. I suggest either reducing the emphasis in the results or discussing the differences in more detail and potentially adding a hypothesis.

General

Over cited paper- reduce the number of citations if they are redundant

Review paper- particularly discussion for redundancies. Some content is repeated with only minor rephrasing.

Flow of introduction/background could improve. Connecting sentences would help connect constructs.

Reviewer #6: This manuscript addresses a relevant topic by linking evolutionary perspectives with current mental health constructs. While the subject is of interest, the conceptual integration between these frameworks would benefit from greater clarity. In particular, the distinction between the impostor phenomenon and self-compassion should be more explicitly defined and theoretically grounded.

The methodology describes analytical parameters, inclusion criteria, and scale validation procedures adapted to the target population. However, the manuscript would benefit from additional detail regarding the sampling strategy, participant characteristics, and the rationale for selecting the measurement instruments, in order to improve transparency and reproducibility.

With respect to the presentation of the results, the formatting of the tables should be improved; specifically, centering the tables would enhance readability and consistency.

Although the results, analysis, and discussion are generally aligned with the reported numerical findings, the interpretation remains largely descriptive. A more critical engagement with the data, including a clearer discussion of variability, limitations, and potential sources of bias, would strengthen the manuscript.

Overall, the study has potential relevance for multiple applied fields, including health, education, and the social sciences. Nonetheless, further conceptual clarification and methodological refinement are necessary to enhance the rigor and impact of the work.

7. PLOS authors have the option to publish the peer review history of their article (what does this mean?). If published, this will include your full peer review and any attached files.

Reviewer #5: No

Reviewer #6: No

---

## [Author Response · Author response to Decision Letter 5]

12 Mar 2026

Dear editors,

please see the version with figures in the attached file Response to reviewers_rev5FIN.

Response to reviewers

We sincerely thank the reviewers for the careful reading of our manuscript and for the valuable and constructive comments, which helped us improve and clarify the manuscript in several respects.

Reviewer #5: Conceptual contribution

1. What the IPP30 adds conceptually (beyond psychometrics)

Authors’ response:

We addressed this issue in the Research background by adding the conceptual explanation: “The IPP30 conceptualises impostorism as a multidimensional profile with distinct facets, which aligns more closely with Clance’s original description of a constellation of impostor-related tendencies. This facet-level view allows domain-specific theorising and practical targeting (e.g., identifying which components of impostorism are most relevant for intervention), and it enables profile or typology approaches that cannot be derived from a single total score.”

Authors’ response:

2. Why competence doubt is the critical link to isolation and overidentification

The critical link was clarified in the Discussion – Summary of results section. “Competence doubt captures the core self-evaluative uncertainty and fear of failure that is most proximal to self-compassion processes. Isolation and overidentification reflect two mechanisms that plausibly intensify these doubts: social disconnection (I am the only one struggling) and cognitive fusion with negative self-evaluations (ruminative overinvolvement). Consistent with this rationale, competence doubt showed the strongest and most consistent associations with isolation and overidentification in our data, motivating its central role in the regression and path models.”

Authors’ response:

3. How this refines existing models

This comment helped us to emphasise the specificity of our results in the Discussion – Summary of results section: “Generally, our facet-level results suggest that the link between self-compassion and impostorism is not uniform but concentrated in competence doubts and the negative self-compassion dimensions. This pattern refines buffer accounts by highlighting isolation and overidentification as potential processes associated with competence-related impostor doubts.”

4. Path model: implies mechanism and directionality.

Although the authors appropriately note that directional paths are not causal, the interpretation of indirect pathways and the narrative sequencing of variables implies a process model that cannot be supported with cross-sectional data.

Several paths are plausibly reversible or reciprocal, particularly the links between isolation and overidentification, competence doubt and working style, and competence doubt and other-self divergence. The cited literature supports these variables’ covariance but does not establish temporal ordering and the unidirectional paths are not sufficiently justified vs other plausible directionality that are not tested.

I recommend testing and reporting fit comparisons for at least one plausible alternative directional model for each key pathway, and/or providing a stronger justification for why the chosen ordering is privileged over alternatives

As currently presented, the directional arrows risk implying mechanistic interpretation that cannot be supported by the present design.

Authors’ response:

We thank the reviewer for the suggestion to test alternative directional specifications of the relationship between overidentification and isolation. In response, we estimated an alternative model (Model_alt1) in which the direction of this path was reversed. This specification showed a substantially poorer model fit and did not adequately represent the data. Importantly, engaging with this alternative prompted us to reconsider the hypothesised status of this association. Given the reciprocal nature suggested in the literature, we revised the model to represent the overidentification–isolation link as a covariance rather than a directional path. This specification proved both theoretically more appropriate and empirically adequate, yielding a slightly improved model fit (Model_current). Similarly, we tested the association between Common Humanity and Self-Kindness as a covariance. However, this specification was not supported empirically, as some of the correlation and regression coefficients were low and did not provide sufficient empirical support for this representation of the relationship (Model_alt4).

In addition, we tested alternative directional specifications for the remaining key pathways (Models_alt2 and alt3). However, these models likewise showed poorer fit and did not improve the overall representation of the data.

Fit Index χ²/df CFI TLI RMSEA AIC (Our model) And AIC (Saturated model)

Model_previous 2.02 .984 .974 .041 70.25 70.000

Model_alt1 18.67 .72 .55 .172 286.65 70.000

Model_current 1.57 .99 .99 0.31 64.78 70.000

Model_alt2 19.02 .74 .54 .173 274.2 70.000

Model_alt3 21.25 .71 .48 .184 300.97 70.000

Model_alt1

Model_current

Model_alt2

Model_alt3

Model_alt4

5. The choice to only regress competence doubt feels post hoc- justify why the choice was made not to regress other facets of IP. Not including the rest of the IP subscales undermines the use of the IPP30. This is a key to what this study adds to the literature.

Authors’ response:

We thank the reviewer for this important and thoughtful comment. We agree that focusing exclusively on competence doubt without examining the remaining IP facets could raise concerns regarding the full use of the IPP30. In response, we extended the regression analyses to include the other IP subscales and now report the relevant results in the Results and Discussion sections. This additional analysis demonstrates that competence doubt occupies a particularly central role within the network of relationships with self-compassion dimensions, while the remaining IP facets show more limited – but in some cases significant - associations.

6. Gender was analyzed and reported but not well interpreted. This factor was also not included in hypotheses. It felt like an important element in the results but then dropped. I suggest either reducing the emphasis in the results or discussing the differences in more detail and potentially adding a hypothesis.

Authors’ response:

Upon further consideration, we agree that the inclusion of gender was not sufficiently integrated into the theoretical framework, as it was not specified a priori in the hypotheses. To maintain theoretical coherence and avoid overemphasising exploratory findings, we therefore decided to remove the gender-related SEM analysis from the main model and results. However, we kept the analysis of the gender differences in IP and SC for a deeper descriptive frame of our research sample, commenting on them and further research direction of this field in the Discussion.

7. Over cited paper- reduce the number of citations if they are redundant.

Authors’ response:

Following a careful review of the reference list, redundant citations were removed, reducing the total number of references from 108 to 95.

8. Review paper- particularly discussion for redundancies. Some content is repeated with only minor rephrasing.

Authors’ response:

We carefully reviewed the Discussion section for redundancies and reduced repeated content across sections, particularly where similar interpretations of the negative self-compassion subscales and impostor phenomenon were presented with minor rephrasing. The revised Discussion now provides a more concise and clearly structured interpretation of the findings.

9. Flow of introduction/background could improve. Connecting sentences would help connect constructs.

Authors’ response:

We restructured part of the section and added/reformulated connecting sentences to support the flow of the text.

Reviewer #6: This manuscript addresses a relevant topic by linking evolutionary perspectives with current mental health constructs. While the subject is of interest, the conceptual integration between these frameworks would benefit from greater clarity.

1. In particular, the distinction between the impostor phenomenon and self-compassion should be more explicitly defined and theoretically grounded.

Authors’ response:

We thank reviewer 6 for their valuable comments, which improved the clarity of our text and the reproducibility of our findings. In response to the first comment, we revised the Introduction to more explicitly and theoretically delineate the distinction between the impostor phenomenon and self-compassion. Specifically, we clarify that the impostor phenomenon reflects a predominantly negative and performance-contingent self-evaluation characterised by relatively specific cognitive–emotional patterns, whereas self-compassion represents a broader, multifaceted self-attitude that shapes how individuals respond to experiences of failure and inadequacy. (…)

2. The methodology describes analytical parameters, inclusion criteria, and scale validation procedures adapted to the target population. However, the manuscript would benefit from additional detail regarding the sampling strategy, participant characteristics, and the rationale for selecting the measurement instruments, in order to improve transparency and reproducibility.

Authors’ response:

We added details regarding the sampling strategy and participant characteristics (section Procedure). Further participant characteristics were not assessed to ensure anonymity and encourage voluntary participation. We also clarified the rationale for instruments by emphasizing their psychometric quality, availability in validated Czech versions, and their conceptual alignment with the theoretical framework and research design of the study (section Measures).

3. With respect to the presentation of the results, the formatting of the tables should be improved; specifically, centering the tables would enhance readability and consistency.

Authors’ response:

We centred all tables to improve readability and ensure formatting consistency throughout the manuscript.

4. Although the results, analysis, and discussion are generally aligned with the reported numerical findings, the interpretation remains largely descriptive. A more critical engagement with the data, including a clearer discussion of variability, limitations, and potential sources of bias, would strengthen the manuscript.

We thank the reviewer for this helpful comment and for highlighting the need for a more critical engagement with the data. In response, we expanded the interpretation of the results in several parts of the manuscript to provide a more nuanced discussion of the observed patterns.

First, we elaborated on the interpretation of the descriptive statistics. In addition to reporting means and standard deviations, we now discuss the variability of the variables more explicitly and interpret the skewness and kurtosis values.

Second, we expanded the interpretation of the correlation coefficients, discussing their relative magnitude and implications for understanding the specific facets of self-compassion associated with competence-related impostor doubts.

Third, we elaborated on the interpretation of the regression results. In addition to reporting the regression coefficients and their statistical significance, the revised text now also discusses the corresponding confidence intervals and their implications for the precision and stability of the estimated effects.

Finally, we expanded the discussion of study limitations and potential sources of bias. In addition to the previously discussed limitations related to the cross-sectional design, which does not allow causal inferences, and the characteristics of the sample (participants from a single faculty, predominantly students, with limited national diversity and gender imbalance), we added a more explicit discussion of potential methodological biases related to the sampling strategy and the use of self-report and self-reflective measures. In this context, we also address the possible influence of common method variance, which may inflate the observed associations.

We believe that these revisions provide a more critical interpretation of the findings and strengthen the overall discussion of the results.

---

## [Decision Letter · Decision Letter 5]

15 Apr 2026

PONE-D-25-34227R5Self-compassion and the impostor phenomenon: Associations and implicationsPLOS One

Dear Dr. Krejčová,

Thank you for submitting your manuscript to PLOS ONE. After careful consideration, we feel that it has merit but does not fully meet PLOS ONE’s publication criteria as it currently stands. Therefore, we invite you to submit a revised version of the manuscript that addresses the points raised during the review process. Please submit your revised manuscript by May 30 2026 11:59PM. If you will need more time than this to complete your revisions, please reply to this message or contact the journal office at plosone@plos.org. Please include the following items when submitting your revised manuscript:

We look forward to receiving your revised manuscript.

Kind regards,

Diego A. Forero, MD; PhD

Academic Editor

PLOS One

**Journal Requirements:**

**Additional Editor Comments:**

I agree with the reviewer 5 on the need for addressing some minor issues.

Reviewers' comments:

Reviewer's Responses to Questions

**Comments to the Author**

1. If the authors have adequately addressed your comments raised in a previous round of review and you feel that this manuscript is now acceptable for publication, you may indicate that here to bypass the “Comments to the Author” section, enter your conflict of interest statement in the “Confidential to Editor” section, and submit your "Accept" recommendation.

Reviewer #5: All comments have been addressed

Reviewer #6: All comments have been addressed

2. Is the manuscript technically sound, and do the data support the conclusions?

Reviewer #5: Yes

Reviewer #6: Yes

3. Has the statistical analysis been performed appropriately and rigorously? 

Reviewer #5: Yes

Reviewer #6: Yes

4. Have the authors made all data underlying the findings in their manuscript fully available?

Reviewer #5: Yes

Reviewer #6: Yes

5. Is the manuscript presented in an intelligible fashion and written in standard English?

Reviewer #5: Yes

Reviewer #6: Yes

6. Review Comments to the Author

Reviewer #5: Thank you for addressing the previous comments- I can see the amount of work put in to revisions over time. Here are some ideas that you can consider

Table 1 could remove skew kurtosis as they are not a problem in this data. You mentioned the range in the narrative and that is sufficient.

- Combine table 2 and 3 would make it easier to follow the M scores and significance of t-tests.

- I don’t feel table 4 is necessary- just reporting that while many variables significantly departed from normality (common with large samples) the skew/kurtosis values were within acceptable range.

- Report VIF to resolve multicollinearity concerns in the regression. I wondered why the correlations between self-kindness mindfulness and competence doubt were significant but that was lost in regression. This could be shared variance vs a true non-significant finding. Just nuance if you want to add that angel and discuss further.

- One last review to make sure causal language is tempered.

Reviewer #6: It is noted that the authors took into account the observations made by the reviewers, particularly regarding the specificity of the information, the organization, the alignment with the figures, and the results.

I consider that the manuscript is acceptable for publication.

7. PLOS authors have the option to publish the peer review history of their article (what does this mean?). If published, this will include your full peer review and any attached files.

Reviewer #5: No

Reviewer #6: No

---

## [Author Response · Author response to Decision Letter 6]

17 Apr 2026

Response to reviewers

We sincerely thank the reviewers for their supportive comments and suggestions, which have helped us to improve the methodological rigour and clarity of the manuscript.

Reviewer #5:

1. Table 1 could remove skew kurtosis as they are not a problem in this data. You mentioned the range in the narrative and that is sufficient.

Authors’ response:

Thank you for this constructive suggestion. The skewness and kurtosis columns have been removed from Table 1. Please note that this change could not be displayed in tracked changes mode because it involved deleting table columns.

2. Combine table 2 and 3 would make it easier to follow the M scores and significance of t-tests.

Authors’ response:

We agree that combining Tables 2 and 3 improves the clarity and readability of the results in this section, and we have merged the tables accordingly.

3. I don’t feel table 4 is necessary- just reporting that while many variables significantly departed from normality (common with large samples) the skew/kurtosis values were within acceptable range.

Authors’ response:

We agree with this comment, and Table 4 has been removed. The relevant interpretation has been incorporated into the text.

4. Report VIF to resolve multicollinearity concerns in the regression. I wondered why the correlations between self-kindness mindfulness and competence doubt were significant but that was lost in regression. This could be shared variance vs a true non-significant finding. Just nuance if you want to add that angel and discuss further.

Authors’ response:

Thank you for this observation. We examined multicollinearity and found that VIF values ranged from 1.41 to 1.76, indicating no concerns. The discrepancy between the significant correlations and non-significant regression coefficients likely reflects shared variance among predictors rather than multicollinearity.

5. One last review to make sure causal language is tempered.

Authors’ response:

Thank you. Although we had already reduced the use of causal language in previous revisions, we conducted an additional careful review of the manuscript and further revised several phrases that could imply causality.

---

## [Editor Report · Decision Letter 6]

22 Apr 2026

Self-compassion and the impostor phenomenon: Associations and implications

PONE-D-25-34227R6

Dear Dr. Krejčová,

We’re pleased to inform you that your manuscript has been judged scientifically suitable for publication and will be formally accepted for publication once it meets all outstanding technical requirements.

Kind regards,

Diego A. Forero, MD; PhD

Academic Editor

PLOS One
---

## [Editor Report · Acceptance letter]

PONE-D-25-34227R6

PLOS One

Dear Dr. Krejčová,

I'm pleased to inform you that your manuscript has been deemed suitable for publication in PLOS One. Congratulations! Your manuscript is now being handed over to our production team.

Kind regards,

on behalf of

Dr. Diego A. Forero

Academic Editor

PLOS One